# Structure-conserving spontaneous transformations between nanoparticles

K.R. Krishnadas[1,*], Ananya Baksi[1,*], Atanu Ghosh[1,*], Ganapati Natarajan[1] & Thalappil Pradeep[1]

Ambient, structure- and topology-preserving chemical reactions between two archetypal nanoparticles, $Ag_{25}(SR)_{18}$ and $Au_{25}(SR)_{18}$, are presented. Despite their geometric robustness and electronic stability, reactions between them in solution produce alloys, $Ag_mAu_n(SR)_{18}$ ($m+n=25$), keeping their $M_{25}(SR)_{18}$ composition, structure and topology intact. We demonstrate that a mixture of $Ag_{25}(SR)_{18}$ and $Au_{25}(SR)_{18}$ can be transformed to any arbitrary alloy composition, $Ag_mAu_n(SR)_{18}$ ($n=1$–24), merely by controlling the reactant compositions. We capture one of the earliest events of the process, namely the formation of the dianionic adduct, $(Ag_{25}Au_{25}(SR)_{36})^{2-}$, by electrospray ionization mass spectrometry. Molecular docking simulations and density functional theory (DFT) calculations also suggest that metal atom exchanges occur through the formation of an adduct between the two clusters. DFT calculations further confirm that metal atom exchanges are thermodynamically feasible. Such isomorphous transformations between nanoparticles imply that microscopic pieces of matter can be transformed completely to chemically different entities, preserving their structures, at least in the nanometric regime.

[1] Department of Chemistry, DST Unit of Nanoscience (DST UNS) and Thematic Unit of Excellence (TUE), Indian Institute of Technology Madras, Chennai 600036, India. * These authors contributed equally to this work. Correspondence and requests for materials should be addressed to T.P. (email: pradeep@iitm.ac.in).

Ambient chemical transformations between nanoparticles leading to hybrid systems that preserve structure and topology are a challenging and poorly explored area in materials science. Atomically precise nanoparticles of noble metals[1–10], often called nanoclusters, which constitute an exploding discipline in nanomaterials, are promising candidates to explore such transformations because of their well-defined structures and drastic changes in their properties, in comparison to their bulk form, arising due to electronic confinement. Most intensely explored of these properties are optical absorption and emission[11–15]; near infrared emission of some of the clusters and their large quantum yields have resulted in new applications[16–20]. Nanoparticles are generally considered stable and are expected to preserve their structural integrity in solution. Chemical reactions between nanoparticles are rarely investigated[21].

Here we show that two archetypal examples of noble metal nanoclusters, $Au_{25}(SR)_{18}$ (refs 22–27) and $Ag_{25}(SR)_{18}$ (ref. 28), manifest dramatic chemical reactivity. We use mass spectrometry to study these reactions in detail. The reaction proceeds in solution through a series of metal atom exchanges leading to alloy clusters, $Ag_mAu_n(SR)_{18}$ ($m + n = 25$; $n = 1–24$), preserving the $M_{25}(SR)_{18}$ stoichiometry; the values of $m$ and $n$ are determined by the starting concentrations of the reactant clusters. These alloy clusters possess an identical structural framework and topology as that of the reactant clusters, therefore presenting a unique example of nanoparticle reactivity. Such reactions proceed through inter-cluster adducts which form spontaneously on mixing. Molecular docking simulations show that van der Waals (vdW) forces between these clusters play an important role in the initial stages of the reaction, and density functional theory (DFT) calculations suggest that bond formation occurs between the staples of the clusters during the reaction. Structure-conserving reactions between nanomaterials of this kind suggest new possibilities of such transformations of materials in general.

## Results

### Formation of the entire range of $Ag_mAu_n(SR)_{18}$ alloys.
The clusters, $Ag_{25}(DMBT)_{18}$ (I) and $Au_{25}(PET)_{18}$ (II) were prepared and characterized by well-established methods described in Methods section. The ligands PET (2-phenylethanethiol) and DMBT (2,4-dimethylbenzenethiol) (Supplementary Fig. 1a for structures) were chosen for two reasons. Firstly, PET is one of the most commonly used protecting ligands for the $Au_{25}$ core and DMBT is the only ligand known so far to protect the $Ag_{25}$ core. Furthermore, molecular masses of these ligands are equal, allowing easy identification of Ag–Au exchanges between I and II. If the ligands were of unequal masses, exchanges of the ligands themselves (DMBT-PET exchange) and that of metal–ligand fragments ((Ag-DMBT)-(Au-PET) exchange), which also occur[21], would complicate the mass spectrometric measurements (see later). Electrospray ionization mass spectra (ESI MS) of I and II show their characteristic features (Fig. 1a,b). Isotopic patterns of these features are identical to their respective theoretical patterns, as shown in Supplementary Fig. 1b,c. The characteristic ultraviolet/visible (UV/vis) absorption features of I and II confirm their identity (Supplementary Fig. 2). Matrix-assisted laser desorption ionization mass spectra (MALDI MS) were also measured for confirmation (Supplementary Figs 3 and 4). All of these data collectively prove the identity and purity of the samples.

To study the reaction between I and II, various volumes of the solution of II were added to a solution of I and the reaction was monitored by time-dependent UV/vis absorption and mass spectrometric measurements (see Methods section for details) of these solutions. The sample was injected to the mass spectrometer within less than 1 min of mixing the solutions and the data collection was completed within 2 min. The mass spectrum collected within 2 min after the addition of solution of II to solution of I at a molar (I:II) ratio of 0.3:1.0 is presented in Fig. 1c. Features of I and II are prominent in this mass spectrum, but reaction product peaks are seen next to the parent peaks. Inset of Fig. 1c is an expansion of the same spectrum in the region between $m/z$ 5,300 to 7,150 showing the emergence of a series of peaks separated by $m/z$ 89. Each of these peaks in this window show features separated by $m/z$ 1, indicating that these clusters are singly charged species. Mass separation of 89 Da indicates the occurrence of Ag–Au exchange ($M_{Au} - M_{Ag} = 89$ Da) between I and II. The molecular masses of DMBT and PET ligands are

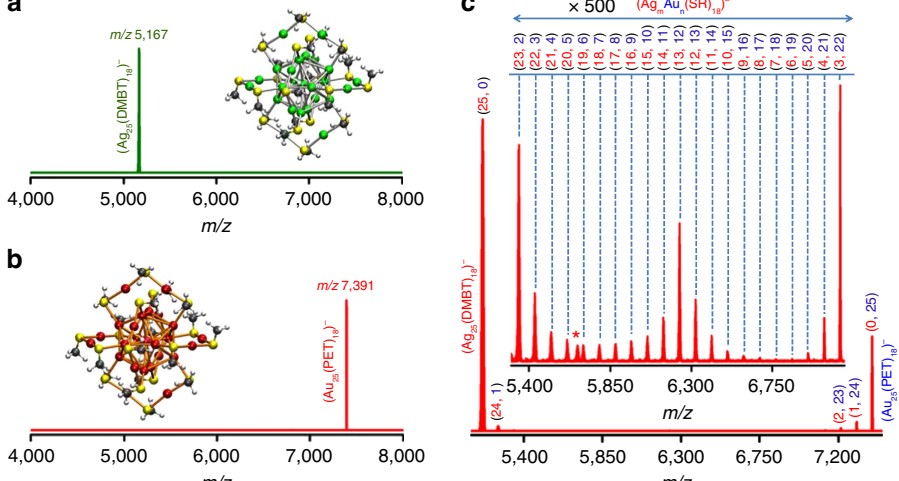

**Figure 1 | Mass spectra of reactant clusters and their entire range of alloys.** Electrospray ionization mass spectra (ESI MS) of $Ag_{25}(DMBT)_{18}$ (**a**), $Au_{25}(PET)_{18}$ (**b**) and a mixture of the two at a $Ag_{25}$:$Au_{25}$ molar ratio of 0.3:1.0 measured within 2 min after mixing (**c**). The peak labels in (**c**) shown as numbers in red ($m$) and blue ($n$) in parentheses give the numbers of Ag and Au atoms, respectively, in the alloy clusters of formula $Ag_mAu_n(SR)_{18}$. Schematic structures of **I** and **II** (with the R group as –$CH_3$, not the real ligands) are also shown. Colour codes for the atoms in the inset pictures: red (Au), green (Ag), yellow (sulfur), black (carbon) and white (hydrogen). The inset in (**c**) is the expanded region of the same mass spectrum between $m/z$ 5,300 to 7,150. **c** and its inset shows that entire range ($n = 1–24$) of alloy clusters, $Ag_mAu_n(SR)_{18}$ ($m + n = 25$), are formed. The peak labelled with '*' in (**c**) is due to an unassigned dianionic species. DMBT is 2,4-dimethylbenzenethiol and PET is 2-phenylethanethiol.

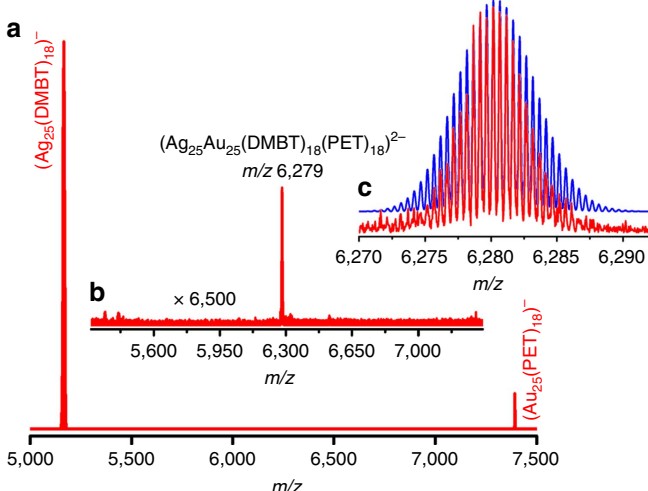

**Figure 2 | Mass spectrometric detection of the intermediate.** Full range electrospray ionization mass spectrum (ESI MS) of the mixture of $Ag_{25}(DMBT)_{18}$ (**I**) and $Au_{25}(PET)_{18}$ (**II**) at molar ratio (**I:II**) of 0.3:1.0 measured within 2 min after mixing (**a**), zoomed in region of panel a in the range between m/z 5,270 and 7,340 showing a feature due to the dianionic adduct, $(Ag_{25}Au_{25}(DMBT)_{18}(PET)_{18})^{2-}$ formed between **I** and **II** (**b**) and theoretical (blue) and experimental (red) isotope patterns of the adduct features (**c**). DMBT is 2,4-dimethylbenzenethiol and PET is 2-phenylethanethiol.

equal (138 Da) and hence the exchanges of ligands (DMBT-PET exchange) and that of metal-ligand fragments ((Ag-DMBT)-(Au-PET) exchange) are not evident from this set of experiments. However, such exchanges were observed in reactions between **I** and $Au_{25}(SBu)_{18}$ (SBu = n-butanethiol) (Supplementary Fig. 5). On the basis of the observation of Ag–Au exchanges, a general formula, $Ag_mAu_n(SR)_{18}$, is given for the alloy clusters formed from **I** and **II**. The numbers in red (m) and those in blue (n) in the parentheses of Fig. 1c represent the numbers of Ag and Au atoms, respectively, in the formula, $Ag_mAu_n(SR)_{18}$. Since the ligand exchange is not detected for these alloys, we do not know the exact numbers of DMBT and PET ligands present in the alloy clusters and hence we use -SR instead of both DMBT and PET separately in the general formula. The total number of metal atoms (25) and that of ligands (18) is preserved in the alloy clusters. Figure 1c also indicates that the entire range of alloy clusters, $(Ag_mAu_n(SR)_{18}$ (n = 1–24; m + n = 25)) that is, $Ag_{24}Au_1(SR)_{18}$ to $Ag_1Au_{24}(SR)_{18}$, are formed in the solution as soon as **I** and **II** are mixed. Another notable observation from Fig. 1c is the higher abundance of $Ag_{13}Au_{12}(SR)_{18}$ (see the peak labelled (13, 12) in Fig. 1c inset) compared with the other alloy clusters. A possible reason for this is discussed in Supplementary Note 1.

**Detection of the intermediate**. We detected the dianionic adduct, $(Ag_{25}Au_{25}(DMBT)_{18}(PET)_{18})^{2-}$, formed between **I** and **II** at a molar ratio (**I:II**) of 0.3:1.0. Instrumental parameters optimized for this set of experiments are listed in Supplementary Notes 2 and 3. Features due to undoped **I** and **II** were observed in the mass spectrum shown in Fig. 2a. The region between m/z 5,270 and 7,340 in this mass spectrum are shown in Fig. 2b. A peak at m/z 6,279 was observed in Fig. 2b which was assigned to $(Ag_{25}Au_{25}(DMBT)_{18}(PET)_{18})^{2-}$. Comparison of the theoretical and experimental isotope patterns shown in Fig. 2c confirms this assignment. A separation of 0.5 mass units between the peaks in

the experimental isotope pattern confirms the dianionic charge state of the adduct. Time-dependent mass spectrometric measurements presented in Supplementary Fig. 6 show that this adduct vanishes almost completely within 5 min and monoanionic alloy cluster products are formed. The prominent features observed at this time are those due to undoped reactant clusters (**I** and **II**). No features due to alloy clusters were detected (except that of $Ag_{24}Au_1(SR)_{18}$, which appears at m/z 5,255; not shown in Fig. 2b).

From Fig. 2a (and from Fig. 1c), we also note that the intensity of **II** is much smaller compared with that of **I** even though higher amounts of **II** are present in the 0.3:1.0 reaction mixture. This could be an indication that some of the **II** might be transformed into mass spectrometrically undetectable or poorly detectable intermediate species during the reaction. For example, anionic $Au_{25}$ can act as an electron donor, which will generate neutral $Au_{25}$, which may not have the same efficiency for mass spectrometric detection (note that we have not used any charge inducing agents such as $Cs^+$ in our measurements to impart charge to neutral species). Furthermore, **II** forms the dianionic adducts with **I** (Fig. 2b and panel a in Supplementary Fig. 6) and also with the alloy clusters formed (panel b in Supplementary Fig. 6). There could be other intermediate species also during the course of the reaction, and probing the details of these events and the intermediates are beyond the scope of the present study. However, these measurements indicate that we have captured one of the earliest events in this reaction, namely the formation of a dianionic adduct of reactant clusters, before the formation of alloys.

Formation of the adduct, $(Ag_{25}Au_{25}(DMBT)_{18}(PET)_{18})^{2-}$, may seem unlikely, considering the overall negative charges of the individual clusters, which may contribute to repulsive interactions between them, preventing their closer approach. However, the charge on these clusters, $Au_{25}(SR)_{18}$ and $Ag_{25}(SR)_{18}$, is not localized but dispersed over atoms in their core, staples and ligands[23,29]. Hence, the two clusters may not experience considerable repulsive interactions between them when they approach each other. Metallophilic interactions[30–34] between the closed-shell Ag(I) and Au(I) centres that are present in the $M_2(SR)_3$ staples of **I** and **II**, and π–π interactions between the aryl groups of the ligands, may also contribute to attractive forces between the two clusters. The sulfur atoms of monothiol ligands (DMBT and PET) involved in these reactions are bound to the metal atoms in the clusters, and these ligands do not have any other functional groups which are free to act as linker molecules between the clusters, unlike in the cases of a few previous reports[35,36]. Moreover, we did not detect the presence of any extra DMBT or PET ligands in the adduct in mass spectrometric measurements, as in the case of clusters having different ligands[37]. These observations suggest that van der Waals (vdW) interactions between the clusters can be one of the factors responsible for adduct formation.

**Nature of the intermediate**. A complete search over the relevant rotational and translational degrees of freedom of one of the clusters w.r.t. the other, and ligand orientational degrees of freedom of each of the two clusters in the adduct with DFT is unfeasible due to the computational cost. Hence, we used a combined approach utilizing the highly efficient force-field-based method of molecular docking to first identify a global minimum energy geometry of the two clusters in close proximity, subject to some constraints (see Methods and Supplementary Note 4 for details of the method) and then optimized this geometry using DFT to yield a more realistic adduct structure. In our initial molecular docking simulations, $Au_{25}(PET)_{18}$ was taken as the

'ligand', that is, the movable molecule whose degrees of freedom would be varied, and $Ag_{25}(DMBT)_{18}$ as the 'receptor' (macromolecule) which was the fixed and completely rigid central molecule. The reason for choosing $Au_{25}(PET)_{18}$ as the movable and flexible molecule was that PET ligands have a greater torsional flexibility than the DMBT ligands due to the longer chain length which would result in lower energy minima during the optimization over the greater number of torsional degrees of freedom. We have used the reported crystal structure coordinates[22,28] of $Ag_{25}(SR)_{18}$ and $Au_{25}(SR)_{18}$, without any structural relaxation, as the input for the molecular docking study. Different types of atoms in the ligands DMBT and PET and the charges on them are given in Supplementary Fig. 7 and Supplementary Tables 1 and 2, respectively.

From the docking simulations, we identified a force-field global minimum geometry (FFGMG) (Supplementary Data 1 for coordinates) of the adduct $(Ag_{25}Au_{25}(DMBT)_{18}(PET)_{18})^{2-}$, as shown in Supplementary Fig. 9. The approach of the two clusters resulted in significant changes in the orientation of the PET ligands (Supplementary Fig. 8 and Fig. 11) at the interface between the clusters so as to permit interdigitation (Supplementary Fig. 10) between the ligands of these clusters.

We then optimized this FFGMG of the adduct using DFT to see if any structural changes might occur due to the additional structural relaxation permitted in DFT (See Methods and Supplementary Note 4; Supplementary Data 2 for coordinates). The exchange-correlation functional employed was the generalized gradient approximation of Perdew, Burke and Ernzerhof (GGA-PBE)[38]. The van der Waals corrections have not been included in this optimization for the reasons explained in Supplementary Note 5. Interestingly, the geometry obtained from the DFT-optimization of the FFGMG of adduct, shown in Fig. 3, reveals that the staples of **I** and **II** have joined through a bond (A–F; 2.90 Å) between a bridging sulfur atom (labelled A) of **II** and an Ag atom (labelled F) of **I**. We note that Ag atom can bind with more than two thiolate ligands as in the case of $Ag_{44}(SR)_{30}$ clusters[10]. Note that the distance between these atoms

in the FFGMG of adduct was longer (3.90 Å) (Supplementary Fig. 9) compared with the A-F bond distance in its DFT-optimized geometry (DFT-OG). A comparison of the FFGMG of the adduct and its DFT-OG shows that the distances between the closest metal and sulfur atoms (Supplementary Fig. 12; Supplementary Note 6) and the ligand orientations (Fig. 3, Supplementary Fig. 9) in these adducts have changed.

A comparison of the average bond distances (Supplementary Table 3) reveals that almost all the bond lengths in both of the clusters in the DFT-OG of adduct have increased significantly compared with the corresponding values in the DFT-OGs of the isolated clusters. For example, average M-sulfur (M = Ag/Au) bond distance in **I** and **II**, in their DFT-OG of adduct, is about 0.1 Å (for **II**) and 0.08 Å (for **I**) longer than the respective values for the DFT-OGs of isolated **I** and **II**. Similarly, average distances of the shorter and longer metal-metal bonds in the icosahedra of **I** (Supplementary Fig. 13 and ref. 39 for description) have also increased in its DFT-OG. Further, the average distance between the central metal atom and the icosahedral surface metal atoms also showed an increase of about 0.04 Å (for **II**) and 0.07 Å (for **I**) in the DFT-OG of adduct. In addition to these changes, the $M_2(SR)_3$ (M = Ag/Au) staples of the two clusters underwent significant changes in terms of their bond angles (Supplementary Fig. 13) as some of the S–M–S (M = Ag/Au) fragments of the staples became more linear compared with their original V-shape in isolated state. In summary, our combined molecular docking simulations and DFT calculations show that formation of the $(Ag_{25}Au_{25}(DMBT)_{18}(PET)_{18})^{2-}$ adduct is feasible during the reaction between **I** and **II**. The predicted adduct structure which features an intercluster Ag–S bond might be one of the initial configurations before further structural and chemical transformations take place in the interfacial region involving the ligands and staple metal atoms.

**Tuning the composition of alloy clusters.** Our experiments showed that the equilibrium distribution of alloy clusters formed is

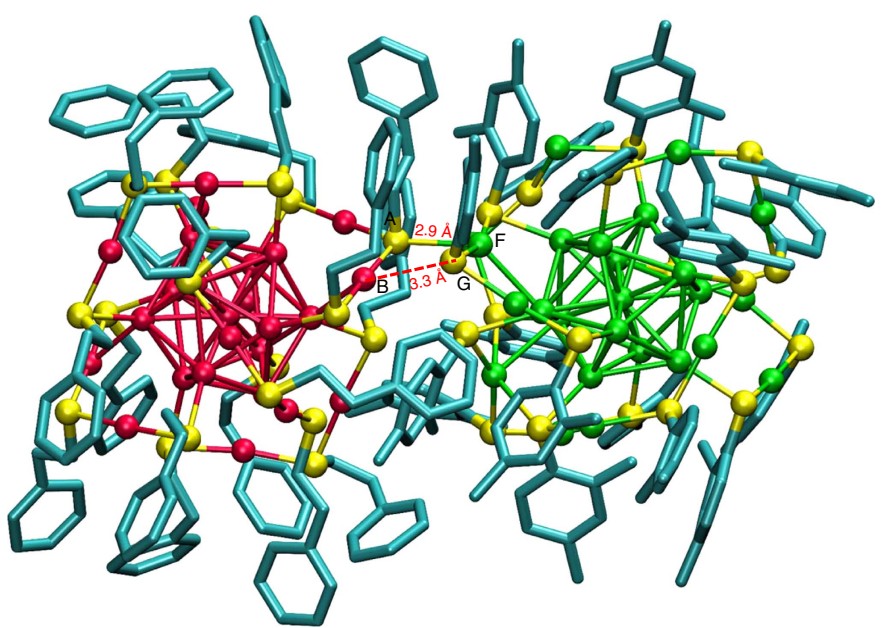

**Figure 3 | DFT-optimized structure of $(Ag_{25}Au_{25}(DMBT)_{18}(PET)_{18})^{2-}$.** The geometry of $(Ag_{25}Au_{25}(DMBT)_{18}(PET)_{18})^{2-}$ adduct (with **II** on the left and **I** on the right) obtained from DFT-optimization of the structure obtained from a force-field-based molecular docking simulation (shown in Supplementary Fig. 9). The hydrogen atoms are omitted from the ligands for clarity. Colour code for the atoms: Au (red), Ag (green), S (yellow) and C (blue). DMBT is 2,4-dimethylbenzenethiol and PET is 2-phenylethanethiol.

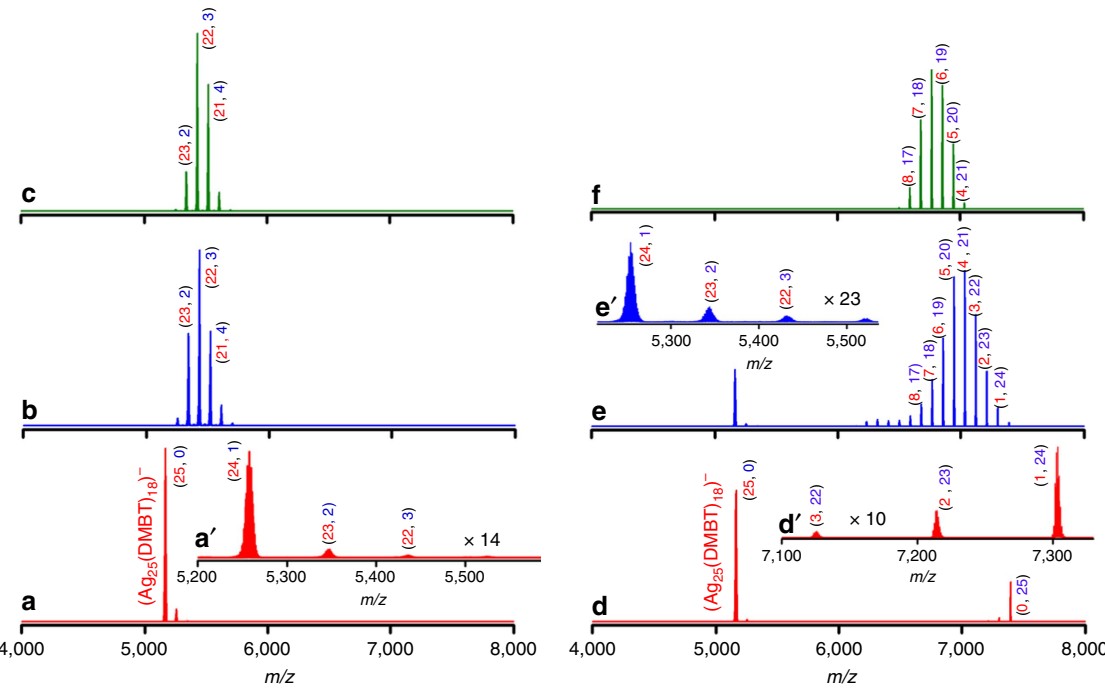

**Figure 4 | Formation of Ag-rich or Au-rich $Ag_mAu_n(SR)_{18}$ ($m+n=25$) clusters.** Electrospray ionization mass spectra (ESI MS) spectra of a mixture of $Ag_{25}(DMBT)_{18}$ (**I**) and $Au_{25}(PET)_{18}$ (**II**) at molar ratios (**I:II**) of 6.6:1.0 (**a–c**) and 0.3:1.0 (**d–f**). DMBT is 2,4-dimethylbenzenethiol and PET is 2-phenylethanethiol. **a–c** and **d–f** correspond to the mass spectra measured within 2 min after mixing, after 10 min and after 1 h, respectively. The peak labels shown as numbers in red (*m*) and blue (*n*) in parentheses give the numbers of Ag and Au atoms, respectively, in the alloy clusters, $Ag_mAu_n(SR)_{18}$. Insets **a′** and **e′** shows the Ag-rich clusters with $n=1–3$. Inset **d′** shows the Au-rich clusters with $n=22–24$.

determined by the relative initial concentrations of the reactant clusters. Figure 4 shows the mass spectra measured at various time intervals during the reaction between **I** and **II** at molar ratios (**I:II**) of 6.6:1.0 (panels **a–c**) and 0.3:1.0 (panels **d–f**), respectively. Temporal changes in the UV/vis absorption features for these reaction mixtures are presented in Supplementary Figs 14 and 15, respectively. Since the concentration of **II** in the reaction mixture was less in the 6.6:1.0 mixture, its feature was not observed in the mass spectrum shown in panel **a** of Fig. 4; however, the doping of Au atoms into **I** (which is in excess) was observed in the spectra measured within 2 min after the addition of **II**, as shown in inset **a′** in Fig. 4. The feature due to undoped **I** disappeared within 10 min and alloy clusters derived from **I** (containing 1–5 Au atoms) appeared, as shown in Fig. 4b. The mass spectrum after 1 h (Fig. 4c) is almost similar to the spectrum after 10 min (Fig. 4b), which shows that no further doping occurred at this composition, even at longer time intervals of the reaction.

To further confirm these observations, we carried out time-dependent MALDI MS measurements (see Supplementary Note 7 for details) for this (6.6:1.0) composition. As shown in Supplementary Fig. 16a, the peak due to undoped **I** and its Au-doped alloys along with their fragments appeared within 2 min after mixing. No features due to **II** and its Ag-doped alloys and their fragments were observed in these measurements. The MALDI MS spectra measured 10 min and 1 h after mixing (Supplementary Fig. 16b,c, respectively) were almost the same and show the formation of Au-doped (containing 1–5 Au atoms) alloys of **I**. Hence, these observations indicate that added **II**, which was lesser in concentration, was consumed completely by **I** present in higher concentration.

When the reaction was carried out at **I:II** molar ratio of 0.3:1.0, features due to the undoped **I** and **II** were observed in the spectra measured within 2 min after mixing, as shown in Fig. 4d. Further,

Ag-doping in **II** was also observed in this spectra (see inset **d′**). Note that the spectra shown in Fig. 4d and in Fig. 1c are the same and these features are described earlier. Mass spectra measured 10 min after mixing (Fig. 4e) show the decrease in intensities of **I** and **II** along with the increase in intensities of Ag-doped **II**. Au-doped alloys of **I** were also observed in this mass spectra (see inset **e′**) but they are significantly lesser in intensity compared with Ag-doped alloys of **II**. After about 1 h, the features due to undoped and Au-doped **I** disappeared completely and only the Ag-doped alloys of **II** (containing 17–21 Au atoms) were observed in the mass spectrum, as shown in Fig. 4f. There were no significant changes in the mass spectra measured at longer time intervals. MALDI MS measurements, presented in Supplementary Fig. 17, further confirm this observation. Alloy clusters with intermediate level of doping (10–14 metal atoms) were observed in a reaction mixture with **I:II** ratio of 1.0:1.0, as shown in Supplementary Figs 18 and 19. The results described above suggest that in the reaction between **I** and **II** at this composition (0.3:1.0), Ag-doping into **II** and Au-doping into **I** were initiated as soon as the two clusters were mixed. However, as the reaction proceeded, the undoped **I** and its Au-doped alloys were consumed by **II** (and its Ag-doped alloys) leading to a mixture of Ag-doped **II**.

The results described above suggest that the cluster which is lesser in concentration in the reaction mixture acts as the source of the dopant metal atom, ligand or metal-ligand fragment, to the cluster higher in concentration. The equilibrium distribution of alloy products for reaction mixtures at various $Ag_{25}:Au_{25}$ molar ratios are presented in Supplementary Fig. 20 which implies that the numbers of Ag and Au atoms in the $Ag_mAu_n(SR)_{18}$ alloy clusters can be continually varied across the entire range ($n=1–24$), simply by varying the initial concentrations of **I** and **II**. We note that attempts to synthesize

Au-doped $Ag_{25}(SR)_{18}$ clusters by galvanic exchange by reaction of $Au^{3+}$ with $Ag_{25}(SR)_{18}$ gave only $Ag_{24}Au_1(SR)_{18}$ (ref. 40), similar to the monosubstituted Pd, Pt and Cd alloys of I and II (refs 41–45). Hence, the reaction between I and II provides a simple method to synthesize alloy clusters $M_{25}(SR)_{18}$ (M = Ag/ Au) with the desired number of Ag and Au atoms.

**Energetics of the exchange reactions.** We carried out DFT calculations (see Methods) to understand the energetics of single Au/Ag atom substitution into the various symmetry unique sites in $M_{25}(SR)_{18}$ and the overall single metal atom substitution reaction, $Ag_{25}(DMBT)_{18} + Au_{25}(PET)_{18} \rightarrow Ag_{24}Au_1(DMBT)_{18} + Au_{24}Ag_1(PET)_{18}$, between I and II. The calculated energies for (1) single Au substitution into $Ag_{25}(DMBT)_{18}$, (2) single Ag substitution into $Au_{25}(PET)_{18}$ and (3) the overall single metal atom substitution reaction are listed in Fig. 5(a–c), respectively. Additional results of the calculations are presented in Supplementary Tables 4–8. Note that the reactions mentioned above do not correspond to any experimental stoichiometry and the reaction energies reported do not give quantitative reaction enthalpies. The energies reported herein do not contain zero-point corrections. However, this method is known to correctly reproduce the experimentally observed site preferences in metal atom substitutions in $M_{25}(SR)_{18}$ systems[46,47]. As shown in Fig. 5d, three distinct sites are available in $Ag_{25}(DMBT)_{18}$ for the first dopant Au atom: these are, (1) the centre of the icosahedron (C), (2) the twelve icosahedral vertex atoms (I) and (3) the twelve staple atoms (S). A similar situation exists for single Ag atom doping into $Au_{25}(PET)_{18}$ since the structure of the two clusters are identical.

Our calculations show that the substitution of Au into all three positions of $Ag_{25}(DMBT)_{18}$ is exothermic. Substitution of Au atom into the centre (C) of $Ag_{25}(DMBT)_{18}$ is the most exothermic (−904 meV) compared with the substitution into I (−540 meV) and S (−578 meV) positions, as shown in Fig. 5(a). This is in contrast to the endothermic substitution of an Ag atom into $Au_{25}(PET)_{18}$ where the C position is the least preferred[46,47], as shown in Fig. 5b. Figure 5c indicates that overall reaction is exothermic for all the combinations of dopant atom locations. Hence, calculations show that exothermicity of the overall

exchange reaction between $Ag_{25}(DMBT)_{18}$ and $Au_{25}(PET)_{18}$ is due to the exothermicities of the metal atom substitution reactions into favourable locations in $Ag_{25}(DMBT)_{18}$ and $Au_{25}(PET)_{18}$.

We think that relative changes in the strengths of various metal–metal bonds in the parent and doped clusters could be one of the factors determining the preference of Ag/Au atoms for the C positions of these clusters. Standard bond dissociation energies (in kJ mol$^{-1}$) for the various bond types present in $Ag_{25}(SR)_{18}$ and $Au_{25}(SR)_{18}$ are: Ag–Ag (162.9 ± 2.9), Au–Au (226.2 ± 0.3), Ag–Au (202.5 ± 9.6) (ref. 48). These values indicate that the order of the bond strengths is: Au–Au > Ag–Ag > Ag–Au. In $M_{25}(SR)_{18}$ (M = Ag/Au) clusters, there are twelve interactions between a metal atom at the central atom (C) position and those in the icosahedral (I) positions. Hence, bonding interactions in $Ag_{24}Au(SR)_{18}$ between the central atom (Au) and icosahedral atoms (Ag) are more preferred due to the significantly increased bonding enthalpy (162.9 – 202.5 = −39.6 kJ mol$^{-1}$) compared with the Ag–Ag interactions in undoped $Ag_{25}(SR)_{18}$. Therefore, the substitution of an Au atom into the C position of $Ag_{25}(SR)_{18}$ would be most exothermic compared with substitution of an Au atom into I and S positions, as shown in Fig. 5(a). Similarly, the substitution of an Ag atom into the C position of $Au_{25}(SR)_{18}$ results in twelve weaker Ag–Au interactions at the cost of the twelve stronger Au–Au interactions, originally present in $Au_{25}(SR)_{18}$. This results in a decrease in bonding enthalpy (226.2 – 202.5 = +23.7 kJ mol$^{-1}$) in $Au_{24}Ag(SR)_{18}$ compared with $Au_{25}(SR)_{18}$. Hence, substitution of an Ag atom into the C position of II is the least preferred compared with those into I and S positions, as shown in Fig. 5b.

**Discussion**

The formation of an adduct indicates that bimolecular events involving the intact clusters occur in solution and such events could be one of the pathways of intercluster reactions. In the light of our results, we suggest that initial binding between I and II during the reaction might occur through (i) the attractive vdW forces between the metal atoms and sulfur atoms, and between the alkyl/aryl groups of ligands and (ii) metallophilic interactions between the metal atoms and sulfur atoms in the staples of the two clusters. These interactions would lead to the weakening of

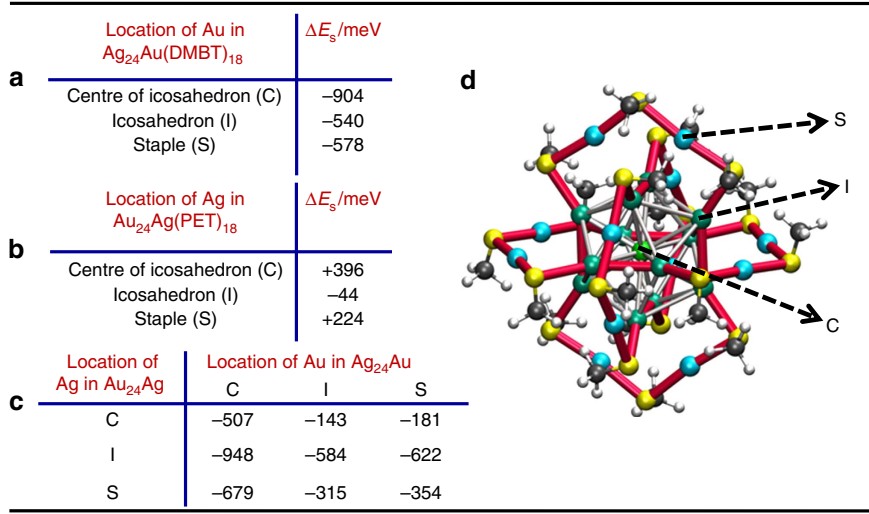

| Location of Au in $Ag_{24}Au(DMBT)_{18}$ | $\Delta E_s$/meV |
|---|---|
| Centre of icosahedron (C) | −904 |
| Icosahedron (I) | −540 |
| Staple (S) | −578 |

**a**

| Location of Ag in $Au_{24}Ag(PET)_{18}$ | $\Delta E_s$/meV |
|---|---|
| Centre of icosahedron (C) | +396 |
| Icosahedron (I) | −44 |
| Staple (S) | +224 |

**b**

| Location of Ag in $Au_{24}Ag$ | Location of Au in $Au_{24}Au$ | | |
|---|---|---|---|
| | C | I | S |
| C | −507 | −143 | −181 |
| I | −948 | −584 | −622 |
| S | −679 | −315 | −354 |

**c**

**Figure 5 | Calculated energies of reactions.** Calculated energies for the substitution reaction ($\Delta E_s$) of Au in $Ag_{25}(DMBT)_{18}$ (**a**), Ag in $Au_{25}(PET)_{18}$ (**b**) and the overall reaction energies (**c**), in meV, as a function of their positions in the product clusters, $Ag_{24}Au_1(DMBT)_{18}$ and $Au_{24}Ag_1(PET)_{18}$. DMBT is 2,4-dimethylbenzenethiol and PET is 2-phenylethanethiol. The overall structure of $M_{25}(SR)_{18}$ (M = Ag/Au) and their distinct metal atom positions (C, I and S) are shown in (**d**). For the labels on the picture, refer to text.

some of the M–S (M = Ag/Au) staple bonds at the interface between the two clusters before the exchange of M, and M–L and L (M = metal, L = ligand) units within the adduct. The DFT-optimized adduct structure shows interesting changes in the structural features of **I** and **II**. One of the most significant pieces of information from our study is the observation of weak bonding between the staples of the two clusters. Such events have been proposed in our earlier study[21]. Further, the interstaple bonding was facilitated by a large number of strained bonds in both the clusters in both core and staple regions, as described earlier. The bonds A–B and G–F (Fig. 3) also might break, leading to staple opening in the subsequent steps of the reaction before exchange of gold and silver atoms. Structural rearrangements in the vicinity of such intercluster bonds could lead to metal and/or ligand exchange.

We note that since the distance between the clusters in the force-field minimum geometry is on the order of 5 Å, electron transfer might occur between these clusters, as such processes have been observed at similar separations for proteins[49]. These processes may generate anionic metal-ligand fragments from one of the clusters, such as $M(SR)_2^-$ (M = Ag/Au), which could attack the $M_2(SR)_3$ staples or the rings[50] of the other cluster, as suggested in our earlier report[21].

The preservation of the structure and topology in the alloy products can be understood from the structural models and the crystal structure data available for $M_{25}(SR)_{18}$ clusters. Based on the divide and protect view[51] of the structure or inherent structure rule[52] proposed for thiolate protected noble metal clusters, these clusters can be viewed as a combination of a symmetrical core of metal atoms surrounded by metal-thiolate units, often referred to as staple motifs. According to these schemes, the $M_{25}(SR)_{18}$ (M = Ag/Au) clusters can be represented as $M@M_{12}(M_2(SR)_3)_6$ where a central atom, M, is surrounded by the $M_{12}$ icosahedron. The resulting $M_{13}$ core can be considered to be protected by six $M_2(SR)_3$ semirings or staple motifs (Fig. 5d). Here, we use the term inherent structure for the basic framework including the geometry of the core (icosahedron, dodecahedron and so on) and that of the staple motifs (M-SR, $M_2(SR)_3$, $M_3(SR)_4$, and so on). The actual shapes of the clusters and slight changes in bond angles and bond lengths, which are essentially dependent on the nature of the element M (Ag/Au) and R groups are not addressed here. Crystal structures of $Ag_{25}(DMBT)_{18}$ and $Au_{25}(PET)_{18}$ show that the inherent structural framework of the icosahedral core and staples, described above, are identical and the presence of different ligand R groups do not affect this structure[22,23,28]. In addition, crystal structure data of alloys have also shown that the substitution of Ag atoms into all the three available positions (C, I and S) in $Au_{25}(SR)_{18}$ does not alter this framework[53,54]. The recently reported structure[40] of $Ag_{24}Au_1(SR)_{18}$ shows that the Au atom occupies the C position, preserving the inherent structure. These results are in agreement with DFT calculations[55] which show that incorporation of twelve Au atoms into the core (or into the staples) of $M_{25}(SR)_{18}$(M = Au/Ag) does not alter the inherent structure. A schematic of the structures of $Au_{13}Ag_{12}(SR)_{18}$ and $Ag_{13}Au_{12}(SR)_{18}$ is presented in Supplementary Fig. 21.

A simplified representation of the inherent structure of $M_{25}(SR)_{18}$ is obtained by the consideration of the ring structures in its bonding network[50]. In this model, these clusters can be represented as $M@(M_8(SR)_6)_3$ where the central atom M is considered to be surrounded by three, interlocked $M_8(SR)_6$ Borromean rings[50,56] (see the path traced by the thick, red bonds in Fig. 5d). This ring representation and the $M@M_{12}(M_2(SR)_3)_6$ model are valid for both $Au_{25}(SR)_{18}$ and $Ag_{25}(SR)_{18}$, and in general for their arbitrary mixed-metal compositions, $Ag_mAu_n(SR)_{18}$, as is evident from their crystallographic data[22,28,53]. In this structural viewpoint, the stability of the inherent structure may be attributed

to the interlocked rings[50,56] and its stiff framework[39]. Hence, such a stable structural configuration would remain intact while atomic units are substituted, involving geometrical distortions and opening and reclosing of parts of the ring structure. Hence, it is the inherent $M@(M_{12})(M_2(SR)_3)_6$ or $M@(M_8(SR)_6)_3$ structure and topology of the bonding network which is preserved during the metal (M), ligand (L) and metal–ligand (M–L) fragment substitution reactions occurring between these clusters.

In conclusion, we report the spontaneous alloying between geometrically robust and electronically stable noble metal clusters $Ag_{25}(SR)_{18}$ and $Au_{25}(SR)_{18}$ in solution, producing $Ag_mAu_n(SR)_{18}$ (n = 1–24; m + n = 25), preserving the inherent structure throughout the entire series. This suggests transformation of a mixture of $Ag_{25}(SR)_{18}$ and $Au_{25}(SR)_{18}$ to $Ag_mAu_n(SR)_{18}$ by successive substitution reactions, which converts one kind of nanoparticle to another, preserving the structure. We detected a dianionic adduct, formed between the two clusters, which could be one of the earliest intermediates in this reaction. Molecular docking simulations combined with DFT calculations show that van der Waals forces as well as bonding between the staple motifs are crucial in forming these adducts. Detection of the dianionic adduct formed between $Ag_{25}(SR)_{18}$ and $Au_{25}(SR)_{18}$ suggests that these clusters themselves, not only their fragments, might be involved in these reactions. This reaction can be used to synthesize bimetallic AgAu clusters, $M_{25}(SR)_{18}$ (M = Ag/Au) clusters with desired composition, simply by adjusting the concentrations of the reactant clusters. DFT calculations reveal that substitutions of Au atoms into all the three symmetry unique sites available in $Ag_{25}(SR)_{18}$ are energetically favourable and exothermic, and this contributes to making the overall reaction exothermic. We hope that our results suggest the prospect of complete transformation of one piece of matter to another, chemically dissimilar one, one atom at a time, preserving structure in the process. Although this has been demonstrated only for two prototypical systems now, the availability of such structures with chemical and structural diversity would enrich this area.

## Methods

**Materials.** The following chemicals were purchased from Sigma Aldrich: Chloroauric acid trihydrate ($HAuCl_4.3H_2O$), 2-phenylethanethiol (PET), n-butanethiol (n-BuS), 2,4-dimethylbenzenethiol (DMBT), tetraoctylammonium bromide (TOAB), tetraphenylphosphonium bromide ($PPh_4Br$) and sodium borohydride ($NaBH_4$). All the solvents used (methanol and dichloromethane (DCM)) were of analytical grade. Silver nitrate ($AgNO_3$) was purchased from RANKEM India.

**Synthesis of clusters.** $Au_{25}(PET)_{18}$ and $Au_{25}(SBu)_{18}$: these clusters were synthesized according to reported methods[21]. $Ag_{25}(SR)_{18}$ was prepared using the procedure of Bakr *et. al.*[28] with slight modifications in the ratio of reagents. Purity of the samples was ensured by ultraviolet/visible (UV/vis) absorption spectroscopy and electrospray ionization mass spectrometry (ESI MS).

**Alloying reaction between $Ag_{25}(SR)_{18}$ and $Au_{25}(SR)_{18}$.** Alloying reaction between these clusters was carried out at room temperature (∼30 °C) by mixing the stock solutions (in DCM) of the two clusters. A fixed volume of the stock solution of $Ag_{25}(DMBT)_{18}$ (**I**) was added into about 1 ml of DCM, and then different volumes of stock solutions of $Au_{25}(PET)_{18}$ (**II**) were added into it, in order to vary the composition of the reaction mixtures. The reaction mixture was not stirred magnetically but mixed using a pipette. Immediate colour changes, time-dependent ultraviolet/vis absorption and ESI MS measurements revealed that the reaction occurred as soon as the two clusters were mixed.

**Instrumentation.** Electrospray ionization mass spectrometric (ESI MS) measurements were carried out using a Waters Synapt G2-Si instrument. ESI MS had a maximum resolution ($m/\Delta m$) of 50,000 in the mass range of interest. We used Applied Biosystems Voyager DEPro mass spectrometer for matrix-assisted laser desorption ionization mass spectrometric (MALDI MS) measurements. More details about the measurements are given in Supplementary Notes 2, 3 and 7. UV/Vis spectra were recorded using a Perkin-Elmer Lambda 25 UV/vis spectrometer. Absorption spectra were typically measured in the range of 200–1100 nm.

**Computational details.** We carried out molecular docking simulations of $Au_{25}(PET)_{18}$ and $Ag_{25}(DMBT)_{18}$ clusters in order to find out how closely they could approach each other and which atoms and their interactions would be involved in the adduct formation. We used the Autodock 4.2 molecular docking software[57] that uses a force-field which includes van der Waals (Lennard-Jones 12-6 potential), hydrogen bonding, desolvation and electrostatics terms and treats the intramolecular bonds and bond angles of both the molecules as rigid, apart from selected bond torsions of the ligands. We used the crystal structure coordinates[22,28] of $Au_{25}(PET)_{18}$ and $Ag_{25}(DMBT)_{18}$, without any structural relaxation, as the input for the docking simulations. Additional details of our methods are given in Supplementary Note 4.

For the DFT-optimization of the force-field global minimum geometry of $(Ag_{25}Au_{25}(DMBT)_{18}(PET)_{18})^{2-}$ obtained from the Autodock program (as described above), we used the real-space grid-based projector augmented wave (GPAW) package[58]. The $Ag(4d^{10}5s^1)$, $Au(5d^{10}6s^1)$, and $S(3s^23p^4)$ electrons were treated as valence and the inner electrons were included in a frozen core. The GPAW setups for gold and silver included scalar-relativistic corrections. The exchange-correlation functional employed was the generalized gradient approximation of Perdew, Burke and Ernzerhof (GGA-PBE)[38]. The van der Waals corrections have not been included in this optimization for the reasons explained in the Supplementary Note 5. We used the LCAO basis set method of GPAW with a 0.2 Å grid spacing for electron density and a convergence criterion of 0.05 eV Å$^{-1}$ for the residual forces on atoms for the structure optimization, without any symmetry constraints. The size of the simulation box was taken to be $40 \times 40 \times 40$ Å$^3$.

For the DFT calculations on the energetics of metal exchange reactions, we adopted the same methods and parameters as described earlier for $(Ag_{25}Au_{25}(DMBT)_{18}(PET)_{18})^{2-}$, except that the more accurate finite-difference method of GPAW was used to compute the total energies of the geometries obtained from an initial optimization using the LCAO basis set method. The crystal structures of $Au_{25}(SR)_{18}$ (ref. 22) and $Ag_{25}(SR)_{18}$ (ref. 28) were used for the initial calculations. The structures of $(Au_{25}(PET)_{18})^-$ and $(Ag_{25}(DMBT)_{18})^-$ were first geometry optimized, and then a single metal (Ag or Au) atom was replaced in a symmetry non-equivalent position and the geometries of the resulting $(Au_{24}Ag(PET)_{18})^-$ and $(Ag_{24}Au(DMBT)_{18})^-$ configurations were then optimized and the energy of the cluster was taken at this energy minimum. We calculated the total energies of Au and Ag atoms using spin-polarization applying Hund's rule to the ground-state electronic configuration of the isolated atoms. The reaction energies of Au/Ag exchange into different positions in both of the clusters were calculated as $E(\text{Reaction}) = E(\text{Products}) - E(\text{Reactants})$.

The structures of **I** and **II**, (with $-CH_3$ instead of PET and DMBT) in Fig. 1a,b were built up with the help of Avogadro software package[59]. We used the coordinates from the crystal structure of $Au_{25}(SR)_{18}$, without any structural relaxation, for building these two structures since the overall structures of these two clusters are the same. The actual structures of $Ag_{25}(DMBT)_{18}$ and $Au_{25}(PET)_{18}$ are not exactly the same due to the arrangement of the ligand -R groups and the bond angles in the ligand shell. All visualizations were created with visual molecular dynamics (VMD) software[60].

**Data availability.** The authors declare that the data supporting the findings of this study are available within the paper and its Supplementary Information Files.

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

## Acknowledgements

K.R.K. and A.G. thank the University Grants Commission for their senior research fellowships. A.B. thanks IIT Madras for an Institute Post Doctoral Fellowship. We thank Department of Science and Technology, Government of India for consistently supporting our research program.

## Author contributions

K.R.K. designed and conducted experiments. A.B. carried out ESI MS measurements. A.G. synthesized the clusters. G.N. carried out DFT calculations and molecular docking simulations. T.P. supervised the whole project. The paper was written by all of the authors.

## Additional information

**Competing financial interests:** The authors declare no competing financial interests.

