## [Peer Review File · Nature Communications]

Reviewers' comments:

Reviewer #1 (Remarks to the Author):

This paper by Pradeep group reports spontaneous alloying between Au₂₅(PET)₁₈ and Ag₂₅(DMBT)₁₈ in solution into Ag_mAu_n(SR)₁₈ while retaining the inherent structure. The new findings are quite convincing because they are based on carefully and intensively conducted experiments and theoretical calculations. It is impressive that the dianionic adducts of Au₂₅(PET)₁₈⁻ and Ag₂₅(DMBT)₁₈⁻ were detected by mass spectrometry at the early stage of the mixing. The surprising chemistry reported here not only provides a novel synthetic route of the composition-controlled alloy clusters but also gives us a new insight into the dynamic aspect of the stability of the magic clusters. As pointed out by the authors themselves (page 14), the present paper suggests that the metal exchange also occurs between individual clusters in the atomically-homogeneous samples. I recommend this clearly-written paper in Nature Communication after addressing the following minor points.

1. The dianionic adduct [Au₂₅Ag₂₅(PET)₁₈(DMBT)₁₈]²⁻ was detected just after the mixing and disappeared with time. The biggest mystery is why adducts with other compositions [Au_nAg_m(SR)₁₈]²⁻ (n+m=25) were not detected during the randomization process.
2. I guess the dianionic adduct [Au₂₅Ag₂₅(PET)₁₈(DMBT)₁₈]²⁻ could be detected more easily in the form of singly-charged anion complexed with the counter cation. Did the authors detect such species?
3. Is it possible to explain the composition distribution of alloy clusters in Figure 1c? Why was the Ag₁₃Au₁₂(SR)₁₈ cluster formed abundantly?
4. It is not clear to me whether the structures of Au₂₅(SR)₁₈ and Ag₂₅(SR)₁₈ in Fig.1 are based on single crystal XRD or theoretical calculation because R is methyl group.
5. I recommend the authors describe explicitly why PET and DMBT with the identical mass were used in the present study.
6. I think that Figure S16 is quite impressive from the viewpoint of synthesis of new alloy clusters. Please consider the possibility to use this figure in the main text.

Reviewer #2 (Remarks to the Author):

The manuscript reports interesting experimental results for the mixtures of thiolate-protected Ag₂₅ and Au₂₅ nanoclusters. In particular, the important body-of-evidence is provided by mass spectrometry which shows that the nanoclusters transform to alloy-based clusters as a function of time, and that this evolution can be monitored by tuning the nanocluster concentration with respect to each other. I find this main experimental finding original and the experimental methodology appears valid.

However, I do not think that the presentation of the story is appropriate. The text is very hard to read for a general reader due to the fact that many important details are not mentioned in the first place. For example, the DMBT/PET ligands are not defined in the first place (Results) and there is no mention of the corresponding solution in the main text (Methods contains some information, but in the very end). The Supplementary Information is exhaustive and there is relevant data that should have been presented in the main text (images of optical absorption, information in notes). Similar lack of detail is observed in the description of the docking simulations which are not reported in a proper detail in the main text, and the reader has no idea of the level of description. The Discussion section is very speculative and contains little additional insight.

The simulation work has significant flaws. The molecular docking simulations rely on classical force fields, and the quality of results depends critically on the chosen potentials. The authors claim in the Supplementary Note 5 that there is no need to include charges for Au and Ag based on Ref. 6.

However, the corresponding reference presents results where those charges are actually being used. This is very relevant because the Au/Ag atoms are in an oxidised form and differ considerably from the atoms in the metallic cores. The nearby sulphur atoms should have realistic charges as well, but they have been taken from other context (values not specified). I am afraid that these choices bias the results, and the reported binding free energy of -38.81 kcal/mol has no meaning. I think that the obtained diatomic adduct should be cross-checked and validated with a full-scale DFT simulation (including vdW corrections).

The DFT simulations of Au/Ag insertion are not performed appropriately. While the machinery (GPAW) and chosen parameters are alright, the authors have made an over-simplification by using simple -SH groups as representatives of the DMBT and PET side groups (both contain an aromatic group). I wonder this choice because the choice of ligand affects the electronic structure and DFT calculations with full ligands are fully doable for such systems nowadays. Furthermore, the vibrational properties of the clusters have not been computed and the reported energies do not contain zero-point energy corrections. Correspondingly, I do not see that the level of accuracy for energetics is at an acceptable level.

The calculated reaction energies are discussed in terms of dissociation energies of various bond types (Au-Au, Au-Ag, Ag-Ag). I wonder why the authors do not consider possible changes in bond distances and strain effects inside the icosahedral core within this context? These must be relevant as well.

As a conclusion, I do not find that the overall presentation and the quality of the theoretical work (simulations) match the criteria for publication in Nature Communications.

Reviewer #3 (Remarks to the Author):

The authors present an extremely interesting study of alloying between Au₂₅ and Ag₂₅ nanoclusters. It is one of the most interesting studies I have read lately. It begins to address the mechanism of alloying for small nanoclusters as well as larger nanoparticles. It is potentially transformative, both within and beyond the smaller gold-thiolate nanocluster field.

I have only a few things for the authors to address and then I think this article is well suited for Nature Communications.

First, a very small (but helpful) thing. In Figure 4, a1-c1, a', a2-c2, b' in a and b are confusing. Could you find a way to simplify this notation? Perhaps just a, b, c, d, e, f, g... (Note: Figure 1 is very clear - I like the color coding.)

A bigger question (but without an obvious answer):

Why weren't Au₂₅ clusters observed in a1? It is not just a concentration effect. Ag₂₅ clusters are observed in a2. Is it possible that the Au₂₅ clusters enter the 0 charge state and thus do not show up in the MS? (Otherwise, the clusters must not be M₂₅ the entire time.) Otherwise, one should see the presence of more gold-containing clusters... this also appears to be true in Figure 4 a2. But, what component is reduced if the Au₂₅ clusters are oxidized? Even in Figure 1c (the same as 4a2), the intensity of the Au₂₅ peak is much smaller than the Ag₂₅ peak, even though the ratio is 1 to 0.3. Oxidation of the Au₂₅ clusters (and potentially some of the alloy clusters) could complicate the analysis; not all of the clusters may be observed. This point should at least be discussed in the paper.

Smaller questions:

How does the "immediately" in Figure 1 differ from the "immediately" in Figure 2?

Figure S6 seems to suggest that Au₁₂Ag₁₃ and Ag₁₂Au₁₃ species are a little extra stable. This doesn't seem to be discussed much in the paper, except sort of in the DFT section. Might be nice to mention this a bit more. It may be important.

Figure S13 caption: I would not use the term "instantaneously". Instead, clarify that it is <5 min (or < 3 min, or whatever is appropriate) as was done well in the main text.

For S16, provide the approximate amount of time for the reaction corresponding to a-j. Is it all the same? Is it complete? etc.

What about the effects of solvent on stabilizing the dimer vs. monomer energetics?

The M₈(SR)₆ rings seem to needlessly complicate the understanding of the Au₂₅(SR)₁₈ structure.

Check the spacing of words in the DFT section of SI. At least in my PDF, a number of words run together.

It could be very instructive as a follow-up paper to perform a similar analysis but with different mass ligands (similar to S5, but a larger study)

RESPONSES TO REVIEWER'S COMMENTS

We are glad that all the reviewers appreciated the work and recommended it for publication. We took some time to respond to the comments as some additional work was carried out. All the questions have been answered in detail and we hope that the manuscript will be accepted.

REVIEWER 1

Reviewer #1 (Remarks to the Author):

This paper by Pradeep group reports spontaneous alloying between Au₂₅(PET)₁₈ and Ag₂₅(DMBT)₁₈ in solution into Ag_mAu_n(SR)₁₈ while retaining the inherent structure. The new findings are quite convincing because they are based on carefully and intensively conducted experiments and theoretical calculations. It is impressive that the dianionic adducts of Au₂₅(PET)₁₈⁻ and Ag₂₅(DMBT)₁₈⁻ were detected by mass spectrometry at the early stage of the mixing. The surprising chemistry reported here not only provides a novel synthetic route of the composition-controlled alloy clusters but also gives us a new insight into the dynamic aspect of the stability of the magic clusters. As pointed out by the authors themselves (page 14), the present paper suggests that the metal exchange also occurs between individual clusters in the atomically-homogeneous samples. I recommend this clearly-written paper in Nature Communication after addressing the following minor points.

1. The dianionic adduct [Au₂₅Ag₂₅(PET)₁₈(DMBT)₁₈]₂⁻ was detected just after the mixing and disappeared with time. The biggest mystery is why adducts with other compositions [Au_nAg_m(SR)₁₈]₂⁻ (n+m=25) were not detected during the randomization process.
2. I guess the dianionic adduct [Au₂₅Ag₂₅(PET)₁₈(DMBT)₁₈]₂⁻ could be detected more easily in the form of singly-charged anion complexed with the counter cation. Did the authors detect such species?
3. Is it possible to explain the composition distribution of alloy clusters in Figure 1c? Why was the Ag₁₃Au₁₂(SR)₁₈ cluster formed abundantly?
4. It is not clear to me whether the structures of Au₂₅(SR)₁₈ and Ag₂₅(SR)₁₈ in Fig.1 are based on single crystal XRD or theoretical calculation because R is methyl group.
5. I recommend the authors describe explicitly why PET and DMBT with the identical mass were used in the present study.
6. I think that Figure S16 is quite impressive from the viewpoint of synthesis of new alloy clusters. Please consider the possibility to use this figure in the main text.

RESPONSE

We thank the reviewer for the highly appreciative comments. We describe below our responses to each of the comments.

1. We had indeed observed other dianionic adducts, $[\text{Ag}_m\text{Au}_n(\text{SR})_{36}]^{2-}$ ($m+n=50$), along with $[\text{Au}_{25}\text{Ag}_{25}(\text{PET})_{18}(\text{DMBT})_{18}]^{2-}$ as shown in the Figure 1 below. To observe these ions along with monomers, slight optimization of instrumental parameters was required as mentioned in the manuscripts and Supplementary Notes 2 and 3. To avoid complexity in the presented data, we showed only the monomer species. The modified figure has now been included in the revised Supplementary Figure S6.

Figure 1. ESI MS spectra of a reaction mixture at the $\text{Ag}_{25}(\text{DMBT})_{18}:\text{Au}_{25}(\text{PET})_{18}$ molar ratio of 0.3:1.0 measured (a) within 2 min after mixing and (b) 5 minutes after mixing. Panel a shows the peak due to $[\text{Ag}_{25}\text{Au}_{25}(\text{DMBT})_{18}(\text{PET})_{18}]^{2-}$, the adduct formed between $[\text{Ag}_{25}(\text{DMBT})_{18}]^{-}$ and $[\text{Au}_{25}(\text{PET})_{18}]^{-}$. Panel (b) shows that this peak disappeared almost completely within 5 min after

mixing of the clusters and monoanionic alloy clusters are formed. Numbers in the parentheses of the labels of main peaks in (a) and (b) correspond to the general formula, $[\text{Ag}_m\text{Au}_n(\text{SR})_{18}]^-$ ($m+n=25$) (for monoanionic alloys). Numbers in the parenthesis of the labels of the peaks marked with * in panel (b) correspond to the general formula, $[\text{Ag}_m\text{Au}_n(\text{SR})_{36}]^{2-}$ ($m+n=50$) (for the dianionic adducts). Some of these ions exist in (a) also.

2. We have studied the ESI MS in detail at high mass range (up to 32 kDa) but we have not seen any such singly-charged anion complexed with any counter cation such as TOAB/PPh₄/Na/Au/Ag, which are likely to be present in the solution.

3. Our DFT calculations show that substitution energy for an Au atom to occupy the I and the S positions of $\text{Ag}_{25}(\text{SR})_{18}$ are almost the same which indicates that Au_{12} can be located in positions I or the S of $\text{Ag}_{25}(\text{SR})_{18}$ with equal probability. Thus substitution of the twelve staple (S)/icosahedral (I) Ag atoms in $\text{Ag}_{25}(\text{SR})_{18}$ by twelve Au atoms produce $\text{Ag}_{13}\text{Au}_{12}(\text{SR})_{18}$. Hence, more of $\text{Ag}_{13}\text{Au}_{12}(\text{SR})_{18}$ could be formed as a result of Au substitution into $\text{Ag}_{25}(\text{SR})_{18}$ on either at the I or at the S positions. Therefore, the probability of formation of $\text{Ag}_{13}\text{Au}_{12}(\text{SR})_{18}$ is higher due to the availability of two types of (I and S) twelve-atom sites for Au atoms.

Further, $\text{Ag}_{13}\text{Au}_{12}(\text{SR})_{18}$ can also be derived from $\text{Au}_{25}(\text{SR})_{18}$ as a result of Ag substitution. Our DFT calculations shows that an Ag atom prefers to occupy the I position rather than the S positions. The complete substitution of all the I positions in $\text{Au}_{25}(\text{SR})_{18}$ by Ag atoms would result in the formation of $\text{Ag}_{12}\text{Au}_{13}(\text{SR})_{18}$. The thirteenth Ag atom can occupy any one of the twelve S positions in the staples (note that C site (centre of icosahedron) is least preferred for Ag atom substitution), resulting in the formation of $\text{Ag}_{13}\text{Au}_{12}(\text{SR})_{18}$. Thus the probability of the formation of $\text{Ag}_{13}\text{Au}_{12}(\text{SR})_{18}$ is higher due to the availability of the twelve staple (S) sites for the thirteenth Ag atom.

We do not think that the abundance of $\text{Ag}_{13}\text{Au}_{12}(\text{SR})_{18}$ is due to any shell closing effects as this abundance is observed only when the concentrations of the reacting clusters are comparable. Though this species was observed with higher abundance immediately after mixing (Figure 1c), and it existed for about 5 min (Supplementary Figure S6), no such species was observed after 1h (panel i of Supplementary Figure S20). Further, Supplementary Figure S20 shows that $\text{Ag}_{13}\text{Au}_{12}$ was not observed with any significantly higher abundance (even at higher concentrations of Au_{25}), in contrast to what is seen in Figure 1c and Supplementary Figure S6. These observations show that significantly higher abundance of $\text{Ag}_{13}\text{Au}_{12}$ is observed only for a few minutes after mixing the clusters. As the reaction proceeds, this species also undergoes further doping. If the observed abundance of $\text{Ag}_{13}\text{Au}_{12}(\text{SR})_{18}$ is due to its higher stability due to any shell closing effects, this species is expected to remain at higher abundance for longer time intervals of the reaction.

In summary, the $\text{Ag}_{13}\text{Au}_{12}(\text{SR})_{18}$ detected can be due to a number of isomers depending on (i) the cluster from which it is derived and (ii) the exact locations of the $\text{Ag}_{12}/\text{Au}_{12}$ and the thirteenth

Ag/Au atom. However, standard mass spectrometry cannot distinguish all the isomers of the formula, $\text{Ag}_{13}\text{Au}_{12}(\text{SR})_{18}$. We think that the abundance of $\text{Ag}_{13}\text{Au}_{12}(\text{SR})_{18}$ could be due to the larger number of ways by which $\text{Ag}_{13}\text{Au}_{12}(\text{SR})_{18}$ can be formed.

4. The structures of $[\text{Au}_{25}(\text{PET})_{18}]^-$ and $[\text{Ag}_{25}(\text{DMBT})_{18}]^-$ were only schematic and they were built up with the help of Avogadro software package and visualizations were created with visual molecular dynamics (VMD) software assuming the coordinates from the crystal structure of $\text{Au}_{25}(\text{SR})_{18}$. Since the overall structure of these two clusters are the same, we have chosen the coordinates of $\text{Au}_{25}(\text{SR})_{18}$ for making the structures of both the clusters shown in Figure 1a and Figure 1b. This has been mentioned in the Computational Details of the manuscript. The $-\text{CH}_3$ group was chosen as the $-\text{R}$ group for a clearer representation of the structures in Fig. 1 of the manuscript.

5. The ligands PET (2-phenylethanethiol) and DMBT (2,4-dimethylbenzenethiol) were chosen for two reasons: (i) PET is one of the most commonly used ligands of Au_{25} core and DMBT is the only ligand known so far to protect the Ag_{25} core exhibiting well defined mass spectra and (ii) molecular masses of these ligands are equal, allowing easy identification of Ag-Au exchanges between $\text{Ag}_{25}(\text{DMBT})_{18}$ and $\text{Au}_{25}(\text{PET})_{18}$. If the ligands were of unequal masses, exchanges of the ligands themselves (DMBT-PET exchange) and that of metal-ligand fragments ((Ag-DMBT)-(Au-PET) exchange), which might also occur, complicate the mass spectrometric measurements. These points have now been described explicitly in the revised manuscript.

6. We tried to include this as a main figure, but found it impossible to include it along with the other figures because of the page limit of the manuscript. However, if the editor desires, we will add it.

All the changes outlined above have been included in the main text.

REVIEWER 2

Reviewer #2 (Remarks to the Author):

The manuscript reports interesting experimental results for the mixtures of thiolate-protected Ag_{25} and Au_{25} nanoclusters. In particular, the important body-of-evidence is provided by mass spectrometry which shows that the nanoclusters transform to alloy-based clusters as a function of time, and that this evolution can be monitored by tuning the nanocluster concentration with respect to each other. I find this main experimental finding original and the experimental methodology appears valid.

However, I do not think that the presentation of the story is appropriate. The text is very hard to read for a general reader due to the fact that many important details are not mentioned in the first

place. For example, the DMBT/PET ligands are not defined in the first place (Results) and there is no mention of the corresponding solution in the main text (Methods contains some information, but in the very end). The Supplementary Information is exhaustive and there is relevant data that should have been presented in the main text (images of optical absorption, information in notes). Similar lack of detail is observed in the description of the docking simulations which are not reported in a proper detail in the main text, and the reader has no idea of the level of description. The Discussion section is very speculative and contains little additional insight.

The simulation work has significant flaws. The molecular docking simulations rely on classical force fields, and the quality of results depends critically on the chosen potentials. The authors claim in the Supplementary Note 5 that there is no need to include charges for Au and Ag based on Ref. 6. However, the corresponding reference presents results where those charges are actually being used. This is very relevant because the Au/Ag atoms are in an oxidised form and differ considerably from the atoms in the metallic cores. The nearby sulphur atoms should have realistic charges as well, but they have been taken from other context (values not specified). I am afraid that these choices bias the results, and the reported binding free energy of -38.81 kcal/mol has no meaning. I think that the obtained diatomic adduct should be cross-checked and validated with a full-scale DFT simulation (including vdW corrections).

The DFT simulations of Au/Ag insertion are not performed appropriately. While the machinery (GPAW) and chosen parameters are alright, the authors have made an over-simplification by using simple -SH groups as representatives of the DMBT and PET side groups (both contain an aromatic group). I wonder this choice because the choice of ligand affects the electronic structure and DFT calculations with full ligands are fully doable for such systems nowadays. Furthermore, the vibrational properties of the clusters have not been computed and the reported energies do not contain zero-point energy corrections. Correspondingly, I do not see that the level of accuracy for energetics is at an acceptable level.

The calculated reaction energies are discussed in terms of dissociation energies of various bond types (Au-Au, Au-Ag, Ag-Ag). I wonder why the authors do not consider possible changes in bond distances and strain effects inside the icosahedral core within this context? These must be relevant as well.

As a conclusion, I do not find that the overall presentation and the quality of the theoretical work (simulations) match the criteria for publication in Nature Communications.

RESPONSE

We thank the reviewer for the comments.

We have taken care of the points raised by the reviewer in view of making it readable to a general reader.

Details of the ligands, DMBT and PET are now added in the main text itself of the revised manuscript. We have now mentioned clearly that the $\text{Ag}_{25}(\text{DMBT})_{18}$ and $\text{Au}_{25}(\text{PET})_{18}$ are referred to as **I** and **II**, respectively and this makes it clear about the identity of the solutions of these clusters and the sequence in which they are mixed for the reaction. We have tried to incorporate the details such as the optical absorption spectra and other results such as notes in the main text, but found it impossible within the page limit. All the essential details to understand the contents are in the main text.

We have now included a more detailed description of the docking simulations; however, it is difficult to include more details on these methods in the main text keeping the other portions of the text.

In the discussion section we provided the results from the DFT optimization of force-field (molecularly docked) global minimum geometry of $[\text{Ag}_{25}\text{Au}_{25}(\text{DMBT})_{18}(\text{PET})_{18}]^{2-}$ adduct and its significance in the reaction mechanism in view of the known and the recently reported structural models. At this point, more insights about the mechanism is not available and further investigations are in progress to understand the mechanisms of these reactions.

Let us now respond to the other concerns briefly.

Regarding the dimer docking/DFT and vdW correction

Charges for the force field

As per the reviewer's suggestion, we have carried out molecular docking studies on the adduct with charges on all the atoms (Au, Ag, S, C and H). However, the configuration of the adduct changes only in terms of the overall relative orientation of the core and staples, or $\text{M}_8(\text{SR})_6$ rings, and the ligand orientations at the intercluster interface, but the closest distances between atoms of the ligands remain similar to the force-field structure in the previous manuscript. We have therefore replaced the old force-field structure with the new force-field structure which is now in the revised supplementary information.

DFT validation of the adduct:

We have carried out the optimization of the Autodock force-field minimum of the adduct using the PBE functional without van der Waals corrections and found significant changes in the geometry of the core and staple regions in terms of bonds strain and distortions, while there were only some small changes in orientation of the two clusters and some changes in distances between the ligands both within a cluster and between those on adjacent clusters. The appearance of a weak intercluster bond between the Ag atom on a staple of $\text{Ag}_{25}(\text{DMBT})_{18}$ and the S-atom on the staple of $\text{Au}_{25}(\text{PET})_{18}$ cluster shows that the intercluster covalent bonding might occur

between the clusters as an intermediate step in the metal/ligand-exchange reaction. We have shown comparison of the geometries of the force field minimum with its DFT-optimized structure in Supplementary Figures S12 and S13 and found that there are significant distortions in the bond angles in the staples and the core of both clusters which can also be viewed as distortions in the $M_8(SR)_6$ rings.

In order to add van der Waals corrections to the DFT calculation, we tried the TS09 method consisting of partial quantum mechanical corrections to empirical vdW force field terms (Tkatchenko and Scheffler, 2009) as implemented in the GPAW code, but found that it was very inefficient for large systems such as this adduct. GPAW does not support Grimme's empirical vdW correction methods (Grimme, 2004) which are much more efficient. Due to the lack of computational resources (32 CPU Xeon with 128 GB), principally memory, we were not able to perform these calculations using more expensive vdW exchange-correlation functionals either.

However, since the orientations of the ligands in the Autodock force-field minimum in which a vdW potential is explicitly included, is very similar to the orientation of the ligands in the DFT-optimized (with no vdW corrections) structure, we concluded that the addition of vdW corrections to DFT will not make a great deal of difference to the structure of the ligands of the dimer, or in other parts of the structure. Therefore, we have replaced the Autodock force-field geometry in Figure 3 of the manuscript with its DFT-optimized geometry. Besides, our focus is not the dimer intermediate but the reaction leading to metal exchange and that finding does not get affected by the accuracy of the dimer structure.

Use of real ligands versus SH

In order to find out the preference in the metal atom substitution sites in $Au_{25}(SR)_{18}/Ag_{25}(SR)_{18}$, clusters, we adopted a reported method (Walter and Moseler 2009; Guidez et al., 2012). In this method, the difference between the absolute energies of the products (alloy cluster and an isolated metal atom that leaves from the parent cluster) and the reactants (the parent cluster and the isolated metal atom that is being incorporated) is used to obtain the preferred substitution sites. This difference in energy is sometimes called as the “substitutional energy” or “reaction energy” (Walter and Moseler 2009; Guidez et al., 2012). However, this energy does not correspond to the actual, quantitative reaction enthalpies since this method does not consider the whole reaction coordinates/pathways and this procedure is used only to obtain the trend in the substitution sites (we have mentioned this point in the revised manuscript). Also note that for such predictions, at least in the case of $M_{25}(SR)_{18}$ systems ($M=Ag/Au$), $-SH$ groups have been used (instead of the real ligands). However, the predictions based on this method matched well with the experimental results. Hence, we do not expect any difference in the trend in the preferred substitution sites. However, in order to confirm this, we have carried out the DFT simulations of the Ag/Au insertion considering the real ligands and the new results are included in the revised manuscript. The trend or the preference in the metal atom substitution sites were the same as our previous results.

Zero-point energy corrections

We have not considered the vibrational properties and zero point energy (ZPE) corrections for the energy of the clusters and the alloys since we are interested only in obtaining the preference for the metal atom substitution sites. We calculated the ZPE energy using the formula, (summation over all n vibrational normal modes) $ZPE = \sum_{i=0}^n 0.5hc\bar{\nu}_i$, where h =Planck's constant, c =velocity of light and $\bar{\nu}$ = frequency of vibrational normal modes of $Au_{25}(SCH_3)_{18}$, $Ag_{25}(SCH_3)_{18}$ and $Ag_{12}Au_{13}(SCH_3)_{18}$ reported by Tlahuice-Flores et al (Tlahuice-Flores et al., 2013). The calculated ZPE values are given in Table 1 below. Differences in zero point energies between various combinations of these clusters and the per-atom energy corrections are given in Table 2 below. We found that these values are small enough to make any significant changes in the trend in the substitution and reaction energies presented in Table 1 in revised manuscript. Hence, we do not expect ZPE differences to change the magnitude of the substitution and reaction energies presented in Table 1 of the revised manuscript, and the trends in the substituent site preferences for Ag and Au atoms.

We remark that similar trend in site preferences was observed in previous reports (Walter and Moseler 2009; Guidez et al. 2012; Aikens 2008) also where SH or -SMe were used instead of the real ligands and these calculations do not contain any zero point corrections.

Table 1: Clusters and their zero point energies as calculated from the vibrational frequencies given in, Tlahuice-Flores et al., 2013

Cluster	Zero point energy (eV)
$Ag_{25}(SCH_3)_{18}$	19.09768
$Ag_{12}Au_{13}(SCH_3)_{18}$	19.20638
$Au_{25}(SCH_3)_{18}$	19.22353

Table 2: Difference in zero point energies given in Table 1

System	Zero point energy difference (meV)	Per atom corrections in ZPE (meV)
$Ag_{25}(SCH_3)_{18}$ - $Ag_{12}Au_{13}(SCH_3)_{18}$	108.70	8.3615 (108.7/13)
$Au_{25}(SCH_3)_{18}$ - $Ag_{25}(SCH_3)_{18}$	125.85	5.034 (125.85/25)
$Au_{25}(SCH_3)_{18}$ - $Ag_{12}Au_{13}(SCH_3)_{18}$	17.15	1.4291 (17.15/12)

Regarding the use of bond length/strain and bond enthalpy for explaining the calculated substituent site preferences

Our calculated site preferences for Ag substitution into $\text{Au}_{25}(\text{SR})_{18}$ is $\text{I} > \text{S} > \text{C}$ and the preference for Au substitution into $\text{Ag}_{25}(\text{SR})_{18}$ is $\text{C} > \text{S} > \text{I}$. We think that changes in the bond distances in the icosahedral positions cannot explain these site preferences, and hence our analysis based on bond enthalpies provides one valid explanation for (i) the least preference for Ag atom in C position of Au_{25} and (ii) the highest preference for an Au atom in the C position of Ag_{25} .

DFT calculations (Guidez *et al.* 2012; Yamazoe *et al.* 2014) show that when the Ag atom is in the central position of $\text{Au}_{24}\text{Ag}(\text{SR})_{18}$, the average centre-icosahedron metal atom bond decreases only slightly in comparison to those in the unsubstituted $\text{Au}_{25}(\text{SR})_{18}$ (Heaven *et al.*, 2008). Calculations by Aikens have shown that the centre-icosahedron metal atom distance does not depend greatly on the type of element (Ag/Au) (Aikens 2008). Similar is the situation when Ag atoms are present in the S positions, *i.e.* in the staples (Aikens 2008). However, crystal structure data (Kumara *et al.* 2014) show that the substitution of Au atoms on the surface of the icosahedron (I positions) by Ag atoms results in the decrease of the bond distance between the central Au atom and the Ag atoms in the I positions, compared to center-icosahedron metal atom bond distances in $\text{Au}_{25}(\text{SR})_{18}$ (Heaven *et al.* 2008).

Thus, the results discussed above show that the substitution of the central Au atom/icosahedral Au atoms in $\text{Au}_{25}(\text{SR})_{18}$ by Ag atoms make a slight decrease in the center-icosahedral metal atom distances. If this small decrease causes a significant strain in the icosahedron, such strain must be present for both the center- as well as icosahedral surface-substituted alloys and hence, a clear preference of Ag atoms to I positions than C positions (which is experimentally and theoretically established in literature) of $\text{Au}_{25}(\text{SR})_{18}$ cannot be explained. Thus, we think that changes in the bond distances in the icosahedral positions cannot explain the observed significant site preference for Ag atom substitution in $\text{Au}_{25}(\text{SR})_{18}$.

Similarly single crystal data (Bootharaju *et al.* 2016) of $\text{Ag}_{24}\text{Au}(\text{SR})_{18}$ show that the substitution of the central Ag atom by an Au atom does not cause any significant changes in the centre-icosahedral metal atom bond distances. Further, as mentioned before, calculations by Aikens, *et al.*, (Aikens 2008) show that the core-central metal atom distances do not significantly depend on the type of the metal atom (Ag vs Au). These results show that the preference for the Ag/Au atom substitution in $\text{Au}_{25}/\text{Ag}_{25}$ cannot be properly explained by the consideration of the icosahedral bond distances. In any case, fineness of these structural aspects do not affect the general conclusions of our work.

We are aware that our current computational resources do not allow a full scale DFT of the dimer with all the ligands including vdW corrections. Rest of the concerns of the reviewer have been addressed completely. The results reported in the paper and the overall conclusions drawn are not affected by the quality of the calculations on the dimer. We hope the reviewer will find this revised article suitable for publication in Nature Communications.

REVIEWER 3

Reviewer #3 (Remarks to the Author):

The authors present an extremely interesting study of alloying between Au₂₅ and Ag₂₅ nanoclusters. It is one of the most interesting studies I have read lately. It begins to address the mechanism of alloying for small nanoclusters as well as larger nanoparticles. It is potentially transformative, both within and beyond the smaller gold-thiolate nanocluster field.

I have only a few things for the authors to address and then I think this article is well suited for Nature Communications.

First, a very small (but helpful) thing. In Figure 4, a1-c1, a', a2-c2, b' in a and b are confusing. Could you find a way to simplify this notation? Perhaps just a, b, c, d, e, f, g... (Note: Figure 1 is very clear - I like the color coding.)

A bigger question (but without an obvious answer):

Why weren't Au₂₅ clusters observed in a1? It is not just a concentration effect. Ag₂₅ clusters are observed in a2. Is it possible that the Au₂₅ clusters enter the 0 charge state and thus do not show up in the MS? (Otherwise, the clusters must not be M₂₅ the entire time.) Otherwise, one should see the presence of more gold-containing clusters... this also appears to be true in Figure 4 a2. But, what component is reduced if the Au₂₅ clusters are oxidized? Even in Figure 1c (the same as 4a2), the intensity of the Au₂₅ peak is much smaller than the Ag₂₅ peak, even though the ratio is 1 to 0.3. Oxidation of the Au₂₅ clusters (and potentially some of the alloy clusters) could complicate the analysis; not all of the clusters may be observed. This point should at least be discussed in the paper.

Smaller questions:

How does the "immediately" in Figure 1 differ from the "immediately" in Figure 2?

Figure S6 seems to suggest that Au₁₂Ag₁₃ and Ag₁₂Au₁₃ species are a little extra stable. This doesn't seem to be discussed much in the paper, except sort of in the DFT section. Might be nice to mention this a bit more. It may be important.

Figure S13 caption: I would not use the term "instantaneously". Instead, clarify that it is <5 min (or < 3 min, or whatever is appropriate) as was done well in the main text.

For S16, provide the approximate amount of time for the reaction corresponding to a-j. Is it all the same? Is it complete? etc.

What about the effects of solvent on stabilizing the dimer vs. monomer energetics?

The M₈(SR)₆ rings seem to needlessly complicate the understanding of the Au₂₅(SR)₁₈

structure.

Check the spacing of words in the DFT section of SI. At least in my PDF, a number of words run together.

It could be very instructive as a follow-up paper to perform a similar analysis but with different mass ligands (similar to S5, but a larger study)

RESPONSE

We thank the reviewer for the comments.

The notations for the panels in Figure 4 are now simplified as per the reviewer's suggestion and the modified figure is included in the revised manuscript.

Regarding the bigger question

Redox processes might occur between the clusters as indicated by the reviewer. It is difficult to detect such changes as the reaction occurs instantaneously after mixing and at least 2 minutes are required to acquire the ESI mass spectra of the reactant clusters. As mentioned by the reviewer, some of the Au₂₅ clusters might enter into the zero charge state which is not detected by ESI MS. In addition to this, some of the parent anionic Au₂₅ clusters can form the dianionic adducts (i) with Ag₂₅ itself (Panel a in Figure 1 above and panel a in Supplementary Fig. S6) and (ii) with the alloy clusters formed (Panel b in Figure 1 above and panel b in Supplementary Fig. S6). Further, as pointed out by the reviewer, there could be some other intermediate species also (derived from the parent clusters) during the course of the reaction to give final products. Probing these intermediate steps/products is important but beyond the scope of the present work. We think that, the reduction in intensity of the Au₂₅(PET)₁₈⁻ in ESI MS measurements could be due to the combined effects of (i) the redox processes between the clusters, (ii) the formation of the dianionic adducts, and (iii) the formation of intermediate clusters which cannot be detected by mass spectrometry. These aspects have now been mentioned in the revised manuscript.

Regarding smaller questions

The mass spectra in Figure 1 and Figure 2 were measured within 2 min after mixing the two reactant cluster solutions. This has now been mentioned in the revised manuscript.

Regarding abundance of Ag₁₃Au₁₂

Our DFT calculations show that substitution energy for an Au atom to occupy the I and the S positions of Ag₂₅(SR)₁₈ are almost the same which indicates that Au₁₂ can be located in positions I or the S of Ag₂₅(SR)₁₈ with equal probability. Thus substitution of the twelve staple (S)/

icosahedral (I) Ag atoms in $\text{Ag}_{25}(\text{SR})_{18}$ by twelve Au atoms produce $\text{Ag}_{13}\text{Au}_{12}(\text{SR})_{18}$. Hence, more of $\text{Ag}_{13}\text{Au}_{12}(\text{SR})_{18}$ could be formed as a result of Au substitution into $\text{Ag}_{25}(\text{SR})_{18}$ on either at the I or at the S positions. Therefore, the probability of formation of $\text{Ag}_{13}\text{Au}_{12}(\text{SR})_{18}$ is higher due to the availability of two types of (I and S) twelve-atom sites for Au atoms.

Further, $\text{Ag}_{13}\text{Au}_{12}(\text{SR})_{18}$ can also be derived from $\text{Au}_{25}(\text{SR})_{18}$ as a result of Ag substitution. Our DFT calculations shows that an Ag atom prefers to occupy the I position rather than the S positions. The complete substitution of all the I positions in $\text{Au}_{25}(\text{SR})_{18}$ by Ag atoms would result in the formation of $\text{Ag}_{12}\text{Au}_{13}(\text{SR})_{18}$. The thirteenth Ag atom can occupy any one of the twelve S positions in the staples (note that C site (centre of icosahedron) is least preferred for Ag atom substitution), resulting in the formation of $\text{Ag}_{13}\text{Au}_{12}(\text{SR})_{18}$. Thus the probability of the formation of $\text{Ag}_{13}\text{Au}_{12}(\text{SR})_{18}$ is higher due to the availability of the twelve staple (S) sites for the thirteenth Ag atom.

We do not think that the abundance of $\text{Ag}_{13}\text{Au}_{12}(\text{SR})_{18}$ is due to any shell closing effects as this abundance is observed only when the concentrations of the reacting clusters are comparable. Though this species was observed with higher abundance immediately after mixing (Figure 1c), and it existed for about 5 min (Supplementary Figure S6), no such species was observed after 1h (panel i of Supplementary Figure S20). Further, Supplementary Figure S20 shows that $\text{Ag}_{13}\text{Au}_{12}$ was not observed with any significantly higher abundance (even at higher concentrations of Au_{25}), in contrast to what is seen in Figure 1c and Supplementary Figure S6. These observations show that significantly higher abundance of $\text{Ag}_{13}\text{Au}_{12}$ is observed only for a few minutes after mixing the clusters. As the reaction proceeds, this species also undergoes further doping. If the observed abundance of $\text{Ag}_{13}\text{Au}_{12}(\text{SR})_{18}$ is due to its higher stability due to any shell closing effects, this species is expected to remain at higher abundance for longer time intervals of the reaction.

In summary, the $\text{Ag}_{13}\text{Au}_{12}(\text{SR})_{18}$ detected can be due to a number of isomers depending on (i) the cluster from which it is derived and (ii) the exact locations of the $\text{Ag}_{12}/\text{Au}_{12}$ and the thirteenth Ag/Au atom. However, standard mass spectrometry cannot distinguish all the isomers of the formula, $\text{Ag}_{13}\text{Au}_{12}(\text{SR})_{18}$. We think that the abundance of $\text{Ag}_{13}\text{Au}_{12}(\text{SR})_{18}$ could be due to the larger number of ways by which $\text{Ag}_{13}\text{Au}_{12}(\text{SR})_{18}$ can be formed.

The time delay before the measurements is now clearly mentioned in the revised manuscript in connection to Figures S13 and S16.

We have not studied the solvent effects on stabilizing the dimer. This was mainly because of the poor stability of $\text{Ag}_{25}(\text{SR})_{18}$ in solvents such as tetrahydrofuran, toluene, hexane, etc. This cluster was stable mostly in solvents such as dichloromethane, acetonitrile, etc. However, the reaction between the two clusters occurred in solvents such as dichloromethane and acetonitrile wherein both the clusters are stable.

We think the $M_8(SR)_6$ structure of $Au_{25}(SR)_{18}$ is useful as an alternative structural model and especially helpful in understanding intercluster reactions. Hence we have retained the discussion on this aspect in the revised manuscript.

The spacing of words in the DFT section of SI has been checked and necessary corrections have been done to the revised manuscript.

We welcome the suggestion of the reviewer on the possibility of a larger study with clusters having different ligands.

References:

1. Tkatchenko, A. and Scheffler, M. Accurate Molecular Van Der Waals Interactions from Ground-State Electron Density and Free-Atom Reference Data Phys. Rev. Lett. 2009, 102, 073005.
2. Grimme, S. Accurate description of van der Waals complexes by density functional theory including empirical corrections J. Comput. Chem. 2004, 25, 1463-1473.
3. Tlahuice-Flores, A. Normal modes of $Au_{25}(SCH_3)_{18}^-$, $Ag_{12}Au_{13}(SCH_3)_{18}^-$ and $Ag_{25}(SCH_3)_{18}^-$ clusters Molecular Simulation 2013, 39, 428-431.
4. Walter and Moseler Ligand-Protected Gold Alloy Clusters: Doping the Superatom J. Phys. Chem. C 2009, 113, 15834–15837.
5. Guidez *et al.* Effects of Silver Doping on the Geometric and Electronic Structure and Optical Absorption Spectra of the $Au_{25-n}Ag_n(SH)_{18}^-$ ($n = 1, 2, 4, 6, 8, 10, 12$) Bimetallic Nanoclusters J. Phys. Chem. C 2012, 116, 20617–20624.
6. Yamazoe *et al.* Preferential Location of Coinage Metal Dopants ($M = Ag$ or Cu) in $[Au_{25-x}M_x(SC_2H_4Ph)_{18}]^-$ ($x \sim 1$) As Determined by Extended X-ray Absorption Fine Structure and Density Functional Theory Calculations J. Phys. Chem. C 2014, 118, 25284–25290.
7. Heaven *et al.* Crystal Structure of the Gold Nanoparticle $[N(C_8H_{17})_4][Au_{25}(SCH_2CH_2Ph)_{18}]^-$ J. Am. Chem. Soc. 2008, 130, 3754-3755.
8. Aikens, C. M. Origin of Discrete Optical Absorption Spectra of $M_{25}(SH)_{18}^-$ Nanoparticles ($M = Au, Ag$) J. Phys. Chem. C 2008, 112, 19797-19800.
9. Kumara *et al.* X-ray Crystal Structure and Theoretical Analysis of $Au_{25-x}Ag_x(SCH_2CH_2Ph)_{18}^-$ Alloy J. Phys. Chem. Lett. 2014, 5, 461–466.
10. Bootharaju *et al.* Templated Atom-Precise Galvanic Synthesis and Structure Elucidation of a $[Ag_{24}Au(SR)_{18}]^-$ Nanocluster Angew. Chem. Int. Ed. 2016, 55, 922 –926.

REVIEWERS' COMMENTS:

Reviewer #1 (Remarks to the Author):

I found that the paper was revised properly according to the comments and believe that the paper is publishable in Nature Communication in the present form. One minor comment is that the phenomena reported herein are somewhat similar to the pseudomorphic transformation of the nanostructures via ion exchange reactions recently reported in Science (vol.351, 1306). The authors may consider to cite this reference.

Reviewer #2 (Remarks to the Author):

The authors have made significant improvements concerning the quality and presentation of theoretical results. In my first report, I already stated the originality of the interesting finding itself, and found the manuscript potential for Nature Communications. In light of the accommodated changes, I see that the manuscript is now publishable after minor revisions which I list below.

Page 9: Please refrain from using "covalent" bonding with interactions between metal and sulphur atoms. This is repeated on page 15 and the Supplemental Information. The type of bonding is simply chemical. Note that sulphur and metal atoms have opposite partial charges ( ionic component).

Page 9: It should be mentioned explicitly that the DFT-optimized results are for GGA-PBE and without dispersion correction in the main text (and in the methods). The authors could include the corresponding discussion (from reviewer response) in the Supplemental Information.

Page 9: It would be instructive to know the DFT binding energy of the adduct. This could be reported together with the classical FF result.

Page 13 (and Table 1): It is useless to report energies with this precision (4 digits) taking into account approximations in the DFT method. It would be appropriate to note that the energies do not include zero-point energy corrections.

Reviewer #3 (Remarks to the Author):

The authors have an interesting study that I think will be great for Nature Communications. They have taken my previous comments into account. I did notice some things that should be addressed in this version before it is ready, though.

The authors added some information about DFT optimization of their dimer. The structural relaxation they describe is characteristic of relaxation that would occur from a crystal structure geometry or an LDA geometry when one does a relaxation with a GGA functional. I think the initial geometries for this must have been the crystal structure coordinates (or possibly LDA) coordinates, not the GGA coordinates they thought they were starting from. The Computational Details section is not really clear on this, since certain things use the crystal structure coordinates. The authors need to check this thoroughly. Also, providing some M-M and M-S bond lengths for the unoptimized and optimized structures would be good.

As a side note, Ag₂₅ is NOT exactly the same as Au₂₅ due to the arrangement of the ligand R groups and the bond angles in the ligand shell (although I know the authors meant that they were nearly

similar). Just clarify this a bit in the Avogadro section.

The 2.9 Å bond is to a silver atom, which is an important point. Silver can bond to more than two thiolates. I would suggest that the authors briefly include that mention in their paper.

In Figure 2, you might be able to increase the clarity of the figure if b and c were used instead of a' and b' (or perhaps just a' instead of b'). It is not clear why you change letters there.

Spelling errors:

parentheses spelled incorrectly p. 4

observation p. 5 (new text)

Spacing still looks off on the top line of p. 16 and between "thepresence". Again, it could have been my PDF version.

RESPONSE TO REVIEWERS' COMMENTS

We are glad that all the reviewers appreciated the work and recommended it for publication. All the remaining questions have been answered in detail below.

REVIEWER 1

Reviewer #1 (Remarks to the Author):

I found that the paper was revised properly according to the comments and believe that the paper is publishable in Nature Communication in the present form. One minor comment is that the phenomena reported herein are somewhat similar to the pseudomorphic transformation of the nanosstructures via ion exchange reactions recently reported in Science (vol.351, 1306). The authors may consider to cite this reference.

RESPONSE

We thank the reviewer for recommending the paper for publication and for pointing out this reference. The cited reference discusses anion exchange reactions of nanocrystals resulting in pseudomorphic products. Anion exchange reactions in nanocrystals are known over a decade. Our work is concerned with the reactions of atomically precise clusters and is very much different from the reference cited. In view of this, we did not cite this reference.

REVIEWER 2

Reviewer #2 (Remarks to the Author):

The authors have made significant improvements concerning the quality and presentation of theoretical results. In my first report, I already stated the originality of the interesting finding itself, and found the manuscript potential for Nature Communications. In light of the accommodated changes, I see that the manuscript is now publishable after minor revisions which I list below.

Page 9: Please refrain from using "covalent" bonding with interactions between metal and sulphur atoms. This is repeated on page 15 and the Supplemental Information. The type of bonding is simply chemical. Note that sulphur and metal atoms have opposite partial charges ( ionic component).

Page 9: It should mentioned explicitly that the DFT-optimized results are for GGA-PBE and without dispersion correction in the main text (and in the methods). The authors could include

the corresponding discussion (from reviewer response) in the Supplemental Information.

Page 9: It would be instructive to know the DFT binding energy of the adduct. This could be reported together with the classical FF result.

Page 13 (and Table 1): It is useless to report energies with this precision (4 digits) taking into account approximations in the DFT method. It would be appropriate to note that the energies do not include zero-point energy corrections.

RESPONSE

We thank the reviewer for the comments and also for recommending the paper for publication in Nature Communications.

We have now omitted the word “covalent” from the manuscript and the supplementary information in all the discussions on the Ag-S bond between the staples of the two clusters in the DFT-optimized geometry of the adduct.

We have now mentioned explicitly in the manuscript and the supplementary information that the DFT-optimized results are for GGA-PBE and without dispersion correction. Justification for the same is given as Supplementary Note 5 in the revised version.

We have now reported the DFT binding energy of the adduct and a few possible contributions to this energy in Supplementary Note 6.

The energy values in Table 1 (which is now Figure 5 in the revised manuscript) is now corrected to three digits. We have also mentioned in the manuscript that these energies do not contain zero point corrections.

REVIEWER 3

Reviewer #3 (Remarks to the Author):

The authors have an interesting study that I think will be great for Nature Communications. They have taken my previous comments into account. I did notice some things that should be addressed in this version before it is ready, though.

The authors added some information about DFT optimization of their dimer. The structural relaxation they describe is characteristic of relaxation that would occur from a crystal structure geometry or an LDA geometry when one does a relaxation with a GGA functional. I think the initial geometries for this must have been the crystal structure coordinates (or possibly LDA) coordinates, not the GGA coordinates they thought they were starting from. The Computational

Details section is not really clear on this, since certain things use the crystal structure coordinates. The authors need to check this thoroughly. Also, providing some M-M and M-S bond lengths for the unoptimized and optimized structures would be good.

As a side note, Ag₂₅ is NOT exactly the same as Au₂₅ due to the arrangement of the ligand R groups and the bond angles in the ligand shell (although I know the authors meant that they were nearly similar). Just clarify this a bit in the Avogadro section.

The 2.9 Å bond is to a silver atom, which is an important point. Silver can bond to more than two thiolates. I would suggest that the authors briefly include that mention in their paper.

In Figure 2, you might be able to increase the clarity of the figure if b and c were used instead of a' and b' (or perhaps just a' instead of b'). It is not clear why you change letters there.

Spelling errors:

parentheses spelled incorrectly p. 4

observation p. 5 (new text)

Spacing still looks off on the top line of p. 16 and between "thepresence". Again, it could have been my PDF version.

RESPONSE

The Computational Methods section has now been checked thoroughly to clarify the points raised by the reviewer.

In Supplementary Table 3, we compared the DFT-optimized geometries (DFT-OG) of the individual clusters and the adduct. We have now included some of the bond lengths from the reported crystal structures of individual Au₂₅(PET)₁₈ and Ag₂₅(DMBT)₁₈ in Supplementary Table 3 to make the comparison clear. From this Table, it is now clear that the differences in bond lengths are larger between crystal structures of individual clusters and DFT-optimized adduct compared to the differences between DFT-optimized structures of individual clusters and DFT-optimized adduct.

We initially used the crystal structure coordinates of Au₂₅(PET)₁₈ and Ag₂₅(DMBT)₁₈, without any DFT structural relaxation, for the molecular docking simulations. The Autodock program we employed allow only the rotations of the ligands (as shown in Supplementary Figure 8) and no change in the metal-metal and metal-sulfur bond lengths and bond angles is permitted. Hence, these bond lengths and angles are the same for the crystal structures of the individual clusters (Au₂₅(SR)₁₈ and Ag₂₅(SR)₁₈) and for the force-field global minimum geometry (FFGMG) of adduct. Note that we used this FFGMG of the adduct as the input for its DFT-optimization. Therefore, we have not included any additional comparison of bond lengths and angles between the FFGMG and DFT-OG of the adduct.

Slight changes in the actual structures of Au₂₅ and Ag₂₅ have now been more clearly mentioned in the Methods section in the revised manuscript.

We have now mentioned in the revised manuscript that Ag atom can bond to more than two thiolate ligands.

Figure 2 is now changed according to the reviewer's suggestion.

The spelling errors pointed out by the reviewer have now been corrected.

Spacing between the words has now been corrected.